# Microwave-Assisted Rapid Synthesis of NH_4_V_4_O_10_ Layered Oxide: A High Energy Cathode for Aqueous Rechargeable Zinc Ion Batteries

**DOI:** 10.3390/nano11081905

**Published:** 2021-07-24

**Authors:** Seokhun Kim, Vaiyapuri Soundharrajan, Sungjin Kim, Balaji Sambandam, Vinod Mathew, Jang-Yeon Hwang, Jaekook Kim

**Affiliations:** Department of Materials Science and Engineering, Chonnam National University, Gwangju 61186, Korea; sin-tw@nate.com (S.K.); soundharajan.007@gmail.com (V.S.); babichunje1@hanmail.net (S.K.); matbalaji@gmail.com (B.S.); bethelvinod@gmail.com (V.M.); hjy@jnu.ac.kr (J.-Y.H.)

**Keywords:** aqueous batteries, zinc-ion batteries, electrochemistry, electrode materials, ammonium vanadate

## Abstract

Aqueous rechargeable zinc ion batteries (ARZIBs) have gained wide interest in recent years as prospective high power and high energy devices to meet the ever-rising commercial needs for large-scale eco-friendly energy storage applications. The advancement in the development of electrodes, especially cathodes for ARZIB, is faced with hurdles related to the shortage of host materials that support divalent zinc storage. Even the existing materials, mostly based on transition metal compounds, have limitations of poor electrochemical stability, low specific capacity, and hence apparently low specific energies. Herein, NH_4_V_4_O_10_ (NHVO), a layered oxide electrode material with a uniquely mixed morphology of plate and belt-like particles is synthesized by a microwave method utilizing a short reaction time (~0.5 h) for use as a high energy cathode for ARZIB applications. The remarkable electrochemical reversibility of Zn^2+^/H^+^ intercalation in this layered electrode contributes to impressive specific capacity (417 mAh g^−1^ at 0.25 A g^−1^) and high rate performance (170 mAh g^−1^ at 6.4 A g^−1^) with almost 100% Coulombic efficiencies. Further, a very high specific energy of 306 Wh Kg^−1^ at a specific power of 72 W Kg^−1^ was achieved by the ARZIB using the present NHVO cathode. The present study thus facilitates the opportunity for developing high energy ARZIB electrodes even under short reaction time to explore potential materials for safe and sustainable green energy storage devices.

## 1. Introduction

The substantial features of being low-cost and highly safe have projected aqueous rechargeable zinc ion batteries (ARZIBs) for potential implementation in large-scale energy storage devices [1,2,3]. In specific, the abundance of natural zinc combined with the high capacity (~820 mAh g^−1^) and low negative voltage (−0.76 V vs. standard hydrogen electrode (SHE)) of the zinc anode can facilitate the plausible use of ARZIBs as an efficient rechargeable device for smart grid applications [4,5,6]. Continual research efforts are witnessed, in this regard, to present ARZIBs as one of the remedies to the expected severe energy crisis [7,8,9]. In particular, such efforts have resulted in many good discoveries of suitable ARZIB cathodes, mainly based on manganese and vanadium-based materials, with structural and diffusion-related hurdles, however, researchers have identified ways for rectifying those snags. For instance, manganese-based cathodes experience poor cycling capability due to inherent Jahn-Teller distortion-induced manganese dissolution into the electrolyte during repeated cycling [10,11,12]. Tactically, this issue is addressed by ensuring the presence of enough manganese (II) ions in the electrolyte via the inclusion of MnSO_4_ additive in the ZnSO_4_ electrolyte, which results in reaching the reaction equilibrium faster and hence aids in the exceptional cyclability of the cathode [13,14,15,16]. Although, vanadium-based electrode materials have dominated the research activities in the lithium-ion battery application field for more than three decades, their use in ARZIBs was initiated only after 2016 [17,18,19,20,21]. In particular, layered vanadium oxide cathodes have gained wide attention for ARZIB applications due to their high capacity related to the wide operational potential range offered by the varied vanadium oxidation states, very stable cyclability, and extremely high rate performance arising from their structural durability [22,23,24,25]. For example, in 2016 Kundu et al. established a high-capacity bilayered Zn-stabilized V_2_O_5_ cathode for ARZIBs that exhibits a stable Zn electrochemical reversibility delivering 220 mAh g^−1^ specific capacity at 15 C with 80% retention over 1000 cycles [25]. Following this, several layered vanadates with their advantageous open framework structures and the multiple vanadium oxidation states leading to enhanced high capacity and supreme rate performances for prospective ARZIB applications were identified and reported [26,27,28,29,30]. Ammonium vanadium bronze, NH_4_V_4_O_10_ (NHVO) with a monoclinic structure consisting of three V^5+^ and one V^4+^ states (variable oxidation states) has been reported to demonstrate reversible electrochemical Zn-intercalation under prolonged cycling due to its structural stability arising from the strong interaction between polyanionic (V_3_O_8_^−^) and cationic layers by forming H-bonding network of N-H^…^O [17,21]. In addition, the existence of NH_4_^+^-ions as pillars between the vanadium interstitial layers offering increased interlayer distance results in enhanced ion diffusion rate and enriched electronic conductivity which will subdue the volume expansion due to divalent Zn^2+^ intercalation/de-intercalation [31]. For example, the larger interplanar spacing (~9.8 Å) and higher diffusion coefficient (~10^−9^–10^−10^ cm^2^ S^−1^) of NHVO compared to that for NH_4_V_3_O_8_ facilitates high zinc storage properties [32]. Recent first-principle calculations predicted that the lowest migration energy barrier (~0.63 eV) for Zn^2+^ diffusion and hence the feasible intercalation path is along the [010] direction in the interlayer region of the layered NHVO structure [33]. A majority of the reported NHVO cathode has been mostly synthesized by the hydrothermal reaction [33,34,35]. Herein, we present an NH_4_V_4_O_10_ with a unique morphology consisting of plate-type particles and prepared by a microwave reaction for use as a cathode in an ARZIB. In combination with metallic zinc as the negative electrode, the prepared cathode delivers a very high specific capacity of ~417 mAh g^−1^ at 0.25 A g^−1^ and remarkably good cycling performance (more than 75% capacity retention after 1500 consecutive cycles at 2.5 A g^−1^ rate) in the presence of 3M Zn(CF_3_SO_3_)_2_ electrolyte solution. Further, the electrochemical analyses aid in suggesting that the reaction mechanism is based on the reversible electrochemical intercalation of divalent Zn^2+^ and H^+^ ions into the layered NHVO cathode. The results from the present study can expand the understanding of vanadate cathodes for the real-time development of various ARZIB cathodes with electrochemical behaviors analogous for safe and green large-scale energy storage devices.

## 2. Experimental Procedures

NHVO sample was prepared by using the microwave reaction method. In brief, appropriate amounts of ammonium vanadate (NH_4_VO_3_) and oxalic acid (H_2_C_2_O_4·_ 2H_2_O) were initially dissolved in deionized water under stirring. Then, the resulting mixture was filled to 70% volume capacity of a 20 mL quartz tube and sealed. The mixture-contained vial was placed in a microwave reactor (Biotage Initiator Third generation, Biotage, Uppsala, Sweden) and heated to 200 °C for 0.5 h and allowed to cool naturally. The precipitate-containing solution was then washed with distilled water and ethanol several times before vacuum drying in an oven at 70 °C for 8 h.

### 2.1. Structural and Morphological Characterization

A Shimadzu X-ray diffractometer (Shimadzu, Kyoto, Japan) was used to record the powder X-ray powder diffraction (PXRD, Cu Kα radiation, with λ = 1.5406 Å) data for the prepared sample. Field-emission scanning electron microscopy (FE-SEM, Hitachi S-4700, Hitachi, Kerfeld, Germany) was used to study the surface morphology of the sample synthesized by the microwave reaction. Also, field-emission transmission electron microscopy (FE-TEM) analyses were performed using a Tecnai F20 instrument (Philips, Alemlo, The Netherlands) located at the Korea Basic Science Institue (KBSI), Chonnam National University) operating at 200 kV. The particle size distribution was calculated using dynamic light scattering method (DLS, ELS-8000, Otsuka electronics, Osaka, Japan). Prior to analysis, the powder sample was dispersed for 30 min in ethanol to obtain a homogenous solution. The surface area and pore size distribution of the samples were calculated using Brunauer–Emmett–Teller model (BET, BELSORP-mini II, MicrotracMRB, Osaka, Japan).

### 2.2. Electrochemical Characterization

A homogenous slurry was prepared by mixing the active material (80%), Super P carbon (10%), and polyacrylic acid (10%) binder in N-methyl-2-pyrrolidone. The doctor blade technique was followed to achieve a uniform coating of the formed slurry on the stainless steel foil current collector followed by vacuum drying at 80 °C. The coating-contained foil was then pressed between hot (~120 °C) stainless steel twin rollers and cut into circular discs to form the cathode. The active material loading was determined to be in the 1.2–1.6 mg range. A glass-fiber separator soaked in 3 M Zn(CF_3_SO_3_)_2_ electrolyte was sandwiched between the prepared cathode and the zinc metal foil anode in a 2032-type coin cell under open-air conditions and aged for 24 h before electrochemical characterization. The electrochemical discharge/charge experiments were performed using a model 2004H battery testing system (BTS, Nagano Keiki Co., Ltd., Tokyo, Japan) at different current densities in the potential range 1.2–0.2 V vs. Zn^2+^/Zn. Cyclic voltammetry (CV) scans were performed using a potentiostat workstation (PGSTAT302N, Autolab, Metrohm AG, Herisau Switzerland).

## 3. Results and Discussion

The NHVO sample powder was prepared by a facile microwave reaction that lasted for a short period of time (~0.5 h). The use of microwave irradiation in material synthesis is highly efficient in terms of maintaining uniform heating for the entire solution, ensuring reproducibility of materials synthesized due to better control of process parameters and facilitates less energy consumption because the irradiation directly heats the precursor solution and evades the heating of the apparatus [36,37,38]. Microwave irradiation is faster than conventional reactions as high reaction temperatures can be reached within a short time period. The rapid heating can help in accelerating the rate of reaction between the precursors in the solution. Moreover, side products are less formed during microwave irradiation. The structural evolution of the prepared powder was analyzed using PXRD and the results are given in Figure 1a. All the diffraction planes of (001), (200), (002), (110), (111), (003), (−112), (−311), (−312), (−205), (020), (−514) and (603) are well-indexed to the standard monoclinic NH_4_VO_10_ with layered structure (JCPDS NO. 31-0075). As anticipated, the (001) line is the highly intense characteristic plane thereby implying that the growth of the particle most likely occurs in the [001] direction. FE-SEM was used to characterize the particle-size and morphology of the prepared sample and the results indicate that a mixed morphology of plate- and belt-shaped particles of a few hundred nanometers and micrometer dimensions, respectively, are distinguishable, as shown in Figure 1b. The high magnification image in Figure 1c shows a few plate-shaped particles conjoined to form a secondary flower-like structure. Further, the thickness of the plate-shaped particles is roughly estimated to be in the range of a few tens of nanometers. This implies that the unique morphology obtained by microwave reaction is within a short reaction time. To obtain further structural information, the TEM studies of the prepared sample were performed and the results (Figure 2a) confirmed the presence of a uniquely mixed morphology of plate-type and belt-type particles, thereby confirming the FE-SEM results. The high magnification TEM image reveals a portion of a single belt-shaped particle with ambiguous lattice fringes (Figure 2b). It is worth noting here that the crystallinity of the prepared NHVO sample appears to be sensitive to the prolonged exposure of the TEM instrument as the lattice fringes in the portion under the focus of the beam signal gradually began to undergo disintegration or slight amorphization thereby making it complicated to observe clear lattice fringes. However, almost clear lattice fringes are evolved in the high magnified TEM image shown in Figure 2b. The observed lattice fringe width of 0.96 nm corresponds to the characteristic (001) mother plane of the NHVO, as observed from the XRD pattern in Figure 1a. This clearly confirms that the particle growth occurs along the crystallographic [001] direction. In addition, the corresponding selected area electron diffraction (SAED) pattern recorded for the prepared NHVO revealed bright spots corresponding to almost single crystalline characteristics (Figure 2c). Furthermore, the matching of the bright spot from the SAED pattern (Figure 2c) corresponding to the (−205) plane in the XRD pattern (shown in Figure 1a) is also depicted thereby confirming the phase purity of the NHVO prepared by a microwave reaction. Moreover, elemental mapping images clearly reveal the uniform distribution of the corresponding elements throughout the area of study (Figure 2d–h).

The DLS method was used to determine that the particle-size distribution for the NHVO sample. The result shown in Figure 3a confirm that the average particle size was less than 600 nm and corroborated well with the electron microscopy results. BET N_2_ adsorption-desorption analysis to determine the surface area confirmed that the layered NHVO sample demonstrates a type IV isotherm with H4 hysteresis (Figure 3b). Further, the surface area and total pore volume was determined to be 13.34 m^2^ g^−1^ and 0.13 cm^3^ g^−1^, respectively. The existence of H4 hysteresis indicates that the NHVO sample consists of a microporous network with slit pore geometry. The steep increase of relative partial pressure (*P/P_0_*) ~ 0.99 typically corresponds to a uniform particle size distribution in the NHVO sample. The Barrett–Joyner–Halenda (BJH) pore size distribution plot in Figure 3c is clearly indicative of the pore size distribution to be in the micrometre range. These results thus confirm that the determined surface area and porous characteristics can ultimately influence the electrochemical property of the prepared NHVO sample.

Electrochemical Zn-intercalation/de-intercalation behavior into the layered NHVO with a unique morphology was initially examined by cyclic voltammetry at 0.1 mV s^−1^ scan rate in the voltage range of 0.2–1.2 V vs. Zn^2+^/Zn in Zn(CF_3_SO_3_)_2_ electrolyte at ambient conditions and the results are given in Figure 4a. The well-observed cathodic shoulder at 0.91 V and peaks located at 0.6 V and 0.4 V corresponds to the multi-step reduction of vanadium to lower oxidation states (V^5+^↔
V^3+^) due to the intercalation of Zn^2+^ ions into the layered NHVO host [22,39]. In the reverse anodic scan, 0.52, 0.71 and 1.01 V correspond to the continuous Zn^2+^ deintercalation across the layered structure of NHVO. This reveals a multi-step reaction mechanism usually known for vanadium-based cathodes. Interestingly, the cathodic peak around 0.8 V for the cycles can be associated with H^+^/H_2_O insertion, usually observed for vanadium-based cathodes demonstrating both Zn^2+^ and H^+^ insertion [40,41]. From the second scan, there are very slight changes in the peak positions and decreased peak currents, which can be associated with the slightly reduced diffusion pathway for the intercalating ions. However, the CV curves observed for successive cycles, are almost superimposable, thus indicating that the electrochemical reversibility in the cathode is stable. Importantly, it is also possible that the probable role of proton intercalation can be considered for the layered cathode structure. In general, the electrochemical Zn^2+^/H^+^-intercalation process can be influenced by the multi-step complex reaction during the discharge process [42].

The initial two galvanostatic discharge/charge profiles recorded at a medium current density of 0.25 A g^−1^ within the 1.2–0.2 V potential range for the prepared NHVO cathode using a coin-type test cell are provided in Figure 4b. The entire process of intercalation/de-intercalation is represented by the typical S-shaped sloping trend that is indicative of a single-phase reaction corresponding to the solid-solution behavior; the observation being in congruence with the CV results. The initial discharge capacity observed for the present NHVO cathode is 417 mAh g^−1^. This is followed by an initial charge capacity of 374 mAh g^−1^ thereby facilitating a 90% Coulombic efficiency by the layered cathode. However, in the second cycle, almost 100% reversible capacity is achieved by the present NHVO cathode (Figure 4b). Upon further cycling, although stable coulombic efficiency is maintained, the capacity fading related to vanadium dissolution seems to be experienced at a moderate current density of 0.25 A g^−1^, as observed from the gradually decreasing trend of the cycle-life profile shown in Figure 4c. In specific, nearly 67% of the highest initial capacity is sustained after 150 discharge/charge electrochemical cycles. Although higher performances are demonstrated for earlier reported NHVO cathodes, the performances presented here are competitive [32,33]. It is apparent that the advantage presented here is that the phase pure NHVO is successfully prepared under a lower reaction time for useful ARZIB applications.

The prolonged cycling ability of NHVO cathode in Zn(CF_3_SO_3_)_2_ electrolyte at very high current densities was also tested using galvanostatic studies. The resulting long-term cyclability pattern is provided in Figure 5a. To achieve steady capacities, the electrode was cycled for the initial five cycles at 0.2 A g^−1^. Thereafter, the cycling current density was maintained as high as 2.5 A g^−1^. Obviously, the NHVO electrode showed 100% coulombic efficiency with a high reversible capacity of almost 128 mAh g^−1^ even after 1500 cycles. Although the capacity declines considerably under long-term cycling, the problem of vanadium dissolution appears to persist and further studies to stabilize the electrode for prolonged cycling are required. Nevertheless, this result indicates that the present layered cathode can consistently demonstrate electrochemical reversibility even at high current rates. For further confirmation on the structural stability of the NHVO cathode under alternate discharge/charge conditions, the rate performance study at progressively increasing current surges between 0.1 and 6.4 A g^−1^ within the voltage window of 1.2–0.2 V in Zn(CF_3_SO_3_)_2_ electrolyte was studied. At each current density, four discharge/charge cycling of the layered cathode was performed to measure the average zinc storage capacities. After the completion of measuring one set of current densities was tested, the next set of testing current densities was applied reversely, i.e., in the progressively decreasing current surges. The sequence of applying two sets of increasing and decreasing current densities alternatively was continued until 90 discharge/charge cycling of the NHVO cathode was performed. The resulting rate performance provided in Figure 5b shows that the NHVO cathode showcased high durability at all applied current rates. In specific, the electrode shows average discharge capacities of 453, 407, 359, 317, 287, 234 and 170 mAh g^−1^ at 0.1, 0.2, 0.4, 0.8, 1.6, 3.2 and 6.4 A g^−1^, respectively. Upon reversing to progressively decreasing current densities in the next set of rate measurements, there is a very insignificant decrease (<9%) in the average specific capacities compared to those corresponding values registered for the initial set of increasing current densities. More importantly, for the consecutive sets of increasing and decreasing current densities applied, the rate performance response of the NHVO electrode suggests no significant variation in the average specific capacities at each applied current density when compared to the corresponding values recorded in the previous sets. For example, the change in the specific capacities registered at 6.4 A g^−1^ at the 26th (165 mAh g^−1^) and 68th (151 mAh g^−1^) discharge cycles, respectively, for the first and second set of varying current density measurements is relatively less (<9%). Finally, the corresponding discharge/charge profiles for the initial set of progressive current densities (Figure 5c) also indicate that the standard S-type trend is maintained except for the loss in capacity due to the usual difference in the applied current rates. Further examination on the structural stability of the prepared electrode was performed via ex situ XRD studies. The XRD patterns recorded after initial discharge, initial charge and after 1500 discharge cycles, respectively, in comparison to the electrode at OCV are presented in Figure 6. The XRD pattern in the initial discharge state shows slight reduction in the peak intensities for all the characteristic planes. While the plane positions are mostly unaltered, the mother plane of (001) slightly shifts towards higher scanning angles after discharge, as observed from the magnified view of the scanning angle around the mother peak in Figure 6. These observations are mostly triggered by the intercalation of Zn^2+^. Upon charging, all the characteristic planes are observed with increased intensities; albeit the peak intensities are not as high as the parent material at OCV. Further, the (001) mother plane regains its original position. After the 1500th discharge cycle, the mother plane of NHVO disappears (Figure 6) indicating that the layered structure of the active material is disintegrated. While the remaining peaks could be still identified; however, some slight shifts in the respective peak positions are observed. In other words, the reversibility of Zn-intercalation in NHVO is effective during the early cycling and upon prolonged cycling (~over 1000 cycles), the cathode material loses its layered structure. Overall, these results clearly validate the structural stability of the present NHVO electrode prepared by microwave reaction.

To understand the cathode potential of the present NHVO cathode in the ARZIB system, the energy and power densities were determined based on the cathode mass at different current densities and the results are presented as a Ragone plot in Figure 7a. The Zn//NHVO system demonstrated an energy density of 306 Wh Kg^−1^ for a given power density of 72 W Kg^−1^ at 0.1 A g^−1^ current density. Further, the energy loss at high specific power is also relatively less; at the high specific power density values of 2304 and 4608 W Kg^−1^, high energy densities of 157 and 107 Wh Kg^−1^ was achieved, as evidenced from Figure 7a. In comparison to some of the vanadate cathodes including ZnV_2_O_7_ [22], LiV_3_O_8_ [28], Na_2_V_6_O_16_·3H_2_O [43], VS_2_ [30], Na_3_V_2_(PO_4_)_3_ [29], and K_2_V_6_O_16_·2.7 H_2_O [39] reported for the ARZIB system, the performance of the present NHVO cathode is remarkable and better suited for plausible real-time applications.

The interpretation of the electrochemical reaction kinetics from the CV response at various scan rates ranging between 0.1 and 0.5 mV s^−1^ for the present NHVO cathode was performed and the results are presented in Figure 7b–d. The trend of the CV curves at increasing scan rates is mostly preserved, but with gradual broadening in the shape. Hence, the major redox peaks are slightly shifted towards higher or lower voltage, respectively, as shown in Figure 7b. The total current reflects on the overall specific capacity registered for the NHVO electrode. The individual contribution of the surface-controlled capacitive and diffusion-induced processes, respectively, to the overall specific capacity at a specific potential during electrochemical cycling can be determined using the below relation [44,45]:(1)$$i=k_{1}v+k_{2\;}v^{1/2}\;$$
where, *i* represents the measured peak current and *v* refers to the scan rate (mV s^−1^). *k*_1_*v* and *k*_2_*v*^1/2^ represent the capacitive and diffusion-controlled process, respectively, for a specific sweep rate. Accordingly, considering the case for the CV curve recorded at 0.1 mV s^−1^ scan rate, the shaded portion corresponding to ~35% of the overall specific-capacity represents the surface-controlled capacitive process (Figure 7c). In other words, this implies that a dominant diffusion-controlled process influences the present NHVO electrode synthesized by the microwave reaction. Nevertheless, the contribution of from the surface and diffusion-controlled process to the overall specific capacity at different scan rates starting from 0.1 to 0.5 mV s^−1^ are determined and presented in Figure 7d. As anticipated, the plot clearly indicates that the surface-related capacitive process gradually increases with increasing scan rates and is predominant at higher scan rates. This implies that the diffusion reactions highly influence the overall specific capacity at low and medium scan rates.

The electrochemical mechanism for layered vanadium oxide cathodes of ARZIBs, in general, can be explained by a multi-stage intercalation mechanism, as observed from the CV and galvanostatic studies. In fact, the initial study on layered vanadium oxide presented the feasibility of reversible intercalation of pure Zn^2+^ via single and two-phase reactions in a Zn_0.25_V_2_O_5_ cathode [25]. Similarly, our group also reported that a complex mechanism involving more than one phase transition is evident in a LiV_3_O_8_ cathode upon reaction with zinc [28]. However, in the following studies, the role of proton involvement as a co-intercalation ion to Zn^2+^ ion in the process of intercalation in layered vanadium oxide cathodes including NaV_3_O_8_·1.5H_2_O and Zn_0.3_V_2_O_5_·1.5H_2_O was stressed [41,46]. The present study supports this finding that Zn^2+^ and H^+^ ions reversibly intercalate into the NHVO host, as observed from the CV and galvanostatic studies. Overall, the plausible electrochemical reaction for the present NHVO cathode in ARZIB system can be expressed by the following equations:

At the cathode:(2)$$x{\mathrm{Zn}}^{2+}+{\mathrm{NH}}_{4}V_{4}O_{10}+2xe^{-}+yH^{+}+ye^{-}\;↔{\mathrm{Zn}}_{x}H_{y}{\mathrm{NH}}_{4}V_{4}O_{10}$$

At the anode:
(3)$$\mathrm{Zn}↔\;{\mathrm{Zn}}^{2+}+2e^{-}$$

Although it has been widely accepted that Zn^2+^ and H^+^ ions contribute to the intercalation process, the sequence followed by the guest ions during intercalation is hugely debated. While one group of researchers consider that Zn^2+^ and H^+^ ions concomitantly access the cathode host, another group believes that these ions consecutively access into/from the host [47,48]. In this scenario, it is a stiff challenge to identify the exact reaction pathway of the intercalating Zn^2+^ and H^+^ ions and requires further studies. Overall, the present study focuses on the presentation of a layered-type NH_4_V_4_O_10_ cathode with flower-like morphology prepared by a microwave reaction that lasts for a very short duration of ~0.5 h for useful ARZIB applications. This NHVO cathode prepared under short microwave irradiation presents competent or even more impressive electrochemical properties in terms of higher specific capacity and long-term cycling stability than some of the reported counterparts synthesized by the hydrothermal method using a relatively long reaction duration (see Table 1 for comparison) [32,49,50,51,52]. Thus, the present microwave NHVO cathode demonstrates notable zinc storage properties and structural stability for electrochemical reversibility under long-term cycling. However, further tuning of the material in terms of improving the performance, particularly during extreme cycling conditions, is required to suit the demands for real-time application of ARZIBs.

## 4. Conclusions

In summary, a layered NH_4_V_4_O_10_ cathode with flower-like particles was synthesized by a microwave method that avoids the undesirable extension of reaction time. The prepared cathode was effectively utilized to develop a high energy ARZIB system in Zn(CF_3_SO_3_)_2_ electrolyte medium. The NHVO cathode demonstrated impressive reversible specific capacities under cycling (277 mAh g^−1^ with 75% capacity retention after 150 cycles at 0.25 A g^−1^ current density), rapid Zn-storage properties (128 mAh g^−1^ after 1500 cycles at 2.5 A g^−1^) with exceptional coulombic efficiencies of almost 100%. Galvanostatic and potentiodynamic measurements indicated the participation of both Zn^2+^ and H^+^ ions in the electrochemical (de-)intercalation (from)into the layered structure and that the reaction mechanism is reversible. Further, the proposed ARZIB system enclosing the present NHVO electrode exhibits a high energy density of 107 Wh Kg^−1^ under high specific power of 4608 W kg^−1^ based on the cathode mass. These values comparatively better those of commercial Pb-acid (~30 Wh Kg^−1^) and the Ni-Cd (~50 Wh Kg^−1^) batteries thereby demonstrating the prospective use of ARZIBs for real-time applications. It is also worth to mention that further improvements in the NHVO cathode can be attempted by tailoring the material using performance-enhancing strategies of forming composites with conductive materials.

## Figures and Tables

**Figure 1 nanomaterials-11-01905-f001:**
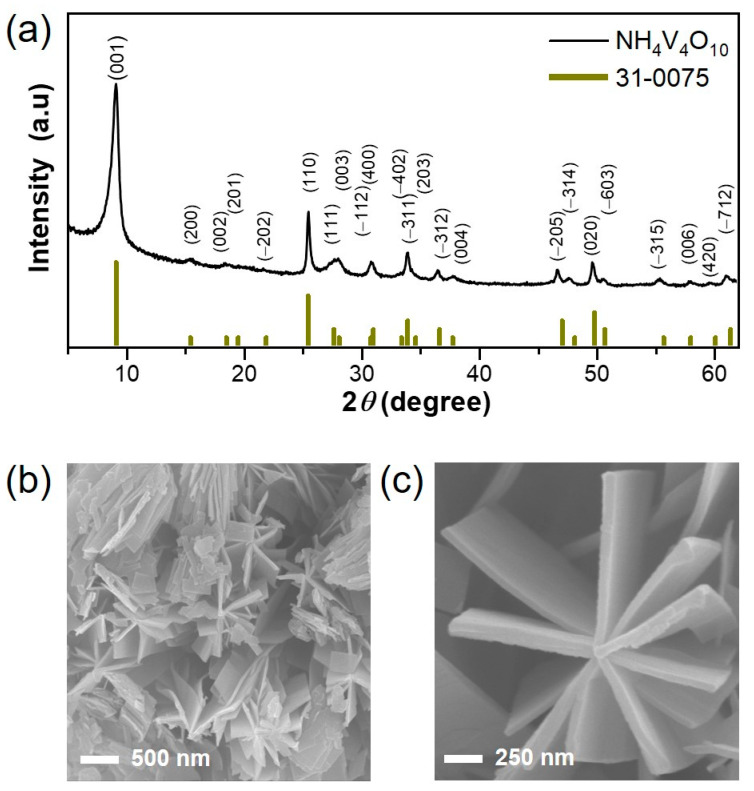
(**a**) Powder XRD pattern of NH_4_V_4_O_10_ (NHVO) cathode prepared by a microwave reaction, (**b**) Low and (**c**) high magnification FE-SEM images of the NHVO electrode.

**Figure 2 nanomaterials-11-01905-f002:**
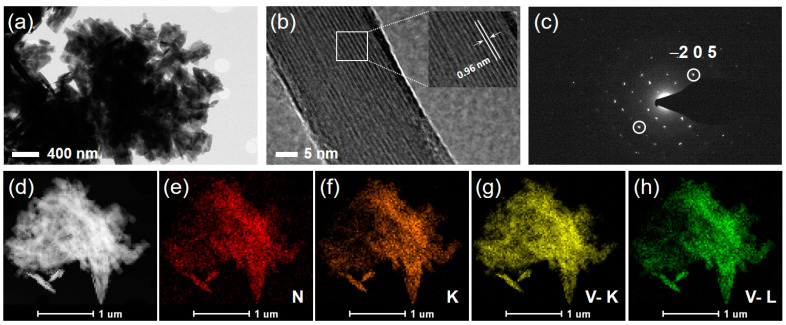
(**a**) Low and (**b**) high resolution TEM images and (**c**) corresponding SAED pattern of the prepared NHVO electrode. (**d**–**h**) FE-TEM and elemental mapping images recorded from the area shown in (**b**). The colored regions correspond to specific elements.

**Figure 3 nanomaterials-11-01905-f003:**
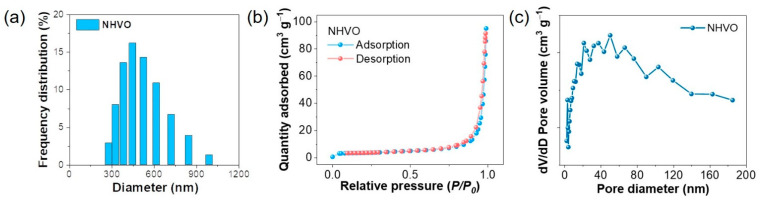
(**a**) DLS result for the prepared NHVO (**b**) N_2_ adsorption/desorption plot and (**c**) corresponding pore-size distribution plot for the prepared NHVO sample.

**Figure 4 nanomaterials-11-01905-f004:**
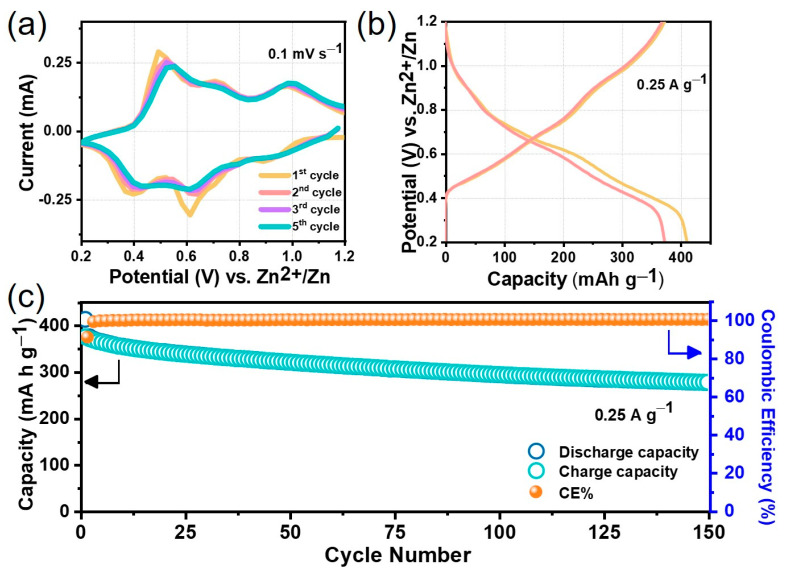
(**a**) The initial five CV profiles of NHVO electrode in ARZIBs, (**b**) Galvanostatic discharge/charge profiles at 250 mA g^−1^ current density for initial two cycles and (**c**) corresponding cyclability configuration of NHVO at the same applied current density.

**Figure 5 nanomaterials-11-01905-f005:**
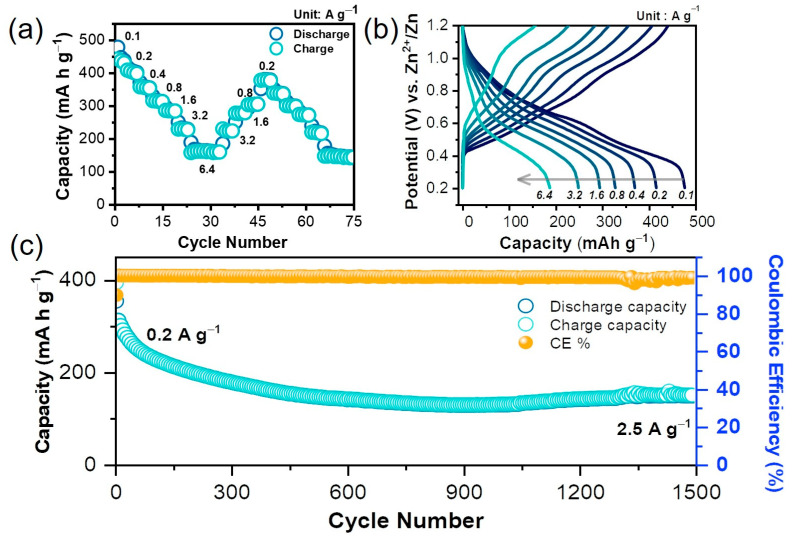
(**a**) Cycle lifespan of the NHVO electrode under prolonged cycling of 1500 cycles at 2.5 A g^−1^ (**b**) Rate performance at different progressively varying (increasing and decreasing alternatively) current densities for the NHVO cathode for ARZIB applications. (**c**) Galvanostatic discharge/charge profiles corresponding to one set of progressively increasing current density from 100 mA g^−1^ to 6.4 A g^−1^.

**Figure 6 nanomaterials-11-01905-f006:**
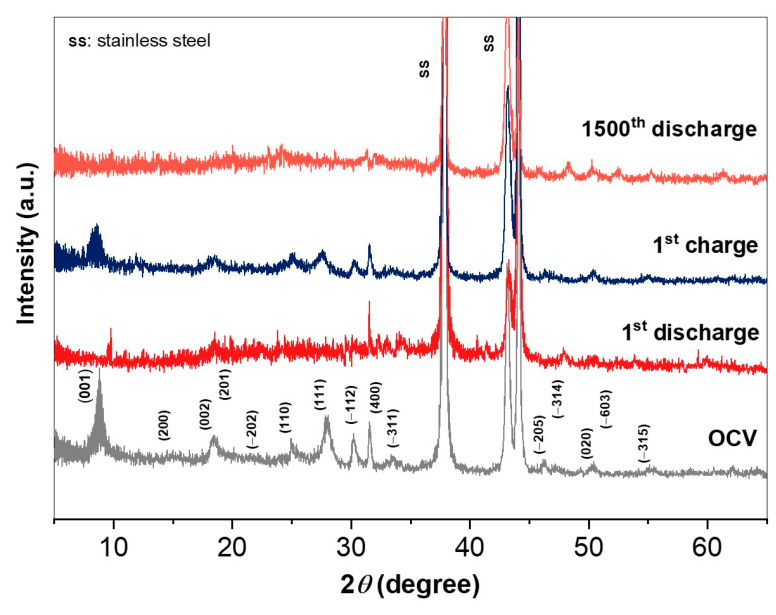
Ex-situ XRD patterns of the NHVO electrode recovered at different conditions.

**Figure 7 nanomaterials-11-01905-f007:**
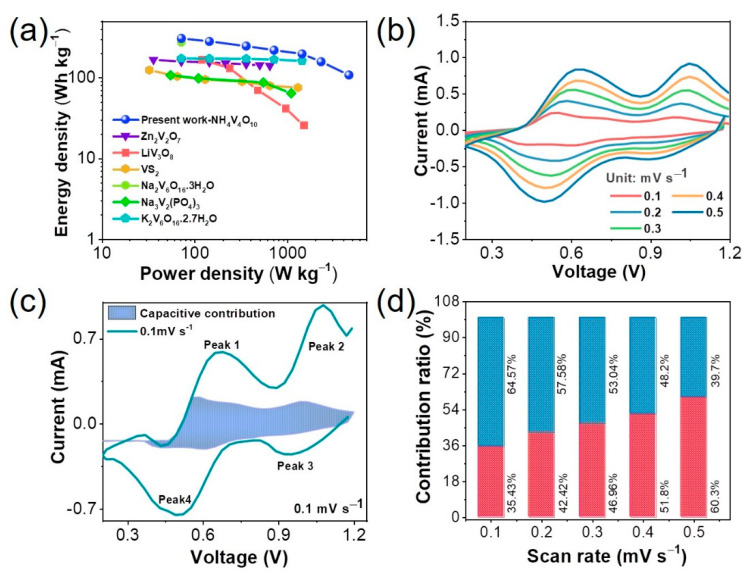
(**a**) Ragone plot for the present NHVO along with reported vanadium-based cathode materials for ARZIBs. (**b**) CV curves at various scan rates between 0.1 and 0.5 mV s^−1^. (**c**) Surface-controlled capacitive contribution (shaded area) to the overall charge storage at 0.1 mV s^–1^. (**d**) Ratio of surface-controlled and diffusion-induced contribution to the charge capacity depicted at different scan rates.

**Table 1 nanomaterials-11-01905-t001:** Comparison of reported ammonium vanadate cathodes and their performance in ARZIBs.

Ammonium Vanadate Based Cathodes	Synthesis Method	Synthesis Time (h)	Maximum Discharge Capacity (mAh g^−1^)/(mA g^−1^)	Rate Capability (mAh g^−1^)/Rate (mA g^−1^)
NH_4_V_4_O_10_ [32]	Hydrothermal	48	~400/300	~180/10,000
Mo-doped NH_4_V_4_O_10_ [49]	Hydrothermal	10	~330/100	~150/2000
NH_4_V_4_O_10_.0.28H_2_O [50]	Hydrothermal	~	~400/200	~112/10,000
NH_4_V_4_O_10-x_.xH_2_O [51]	Hydrothermal	~	~410/50	~120/3000
(NH_4_)_x_V_2_O_5_.nH_2_O [52]	Hydrothermal	48	~370/100	N/A
NH_4_V_4_O_10_ [This work]	Microwave	0.5	~450/100	~170/6400

## Data Availability

The data presented in this study are available on request from the corresponding author.

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
