# Peer review of "Microwave-Assisted Rapid Synthesis of NH4V4O10 Layered Oxide: A High Energy Cathode for Aqueous Rechargeable Zinc Ion Batteries"

_nanomaterials, 2021, doi:10.3390/nano11081905_

Round 1

Reviewer 1 Report

The paper is interesting and relates with a hot topic: aqueous rechargeable zinc ion batteries.  I recommend that the paper is accepted in its present form, provided that the authors take into account the following minor observations/suggestions:

-explain the all the acronyms in text as well, not only in the abstract (e.g. ARZIB, NHVO) when they are used for the first time;

-there are some typos, missing/more spaces, or higher characters than the others (e.g. page 4, line 148: Fig.1).

Reviewer 2 Report

The manuscript in the current form is acceptable. Some suggestions to improve the quality of the paper is as follows. The authors are encouraged to measure the surface area of NH4V4O10 layered oxide and report in the manuscript. It worth to measure the impedance of the cells at different stages of charge and discharge and calculate the diffusion coefficient based on EIS and compare with the data collected by CV tests.
